# Required Visibility Level for Reliable Object Detection during Nighttime Road Traffic in Non-Urban Areas

**Anil Erkan** *[ID], **David Hoffmann** [ID], **Nikolai Kreß, Tsoni Vitkov, Korbinian Kunst** [ID], **Markus Alexander Peier** [ID] **and Tran Quoc Khanh** [ID]

Laboratory of Adaptive Lighting Systems and Visual Processing, Technical University of Darmstadt, Hochschulstr. 4a, 64289 Darmstadt, Germany

* Correspondence: erkan@lichttechnik.tu-darmstadt.de; Tel.: +49-6151-16-22884

**Abstract:** Motor vehicle headlamps are the only light sources that create visibility conditions for the driver in nighttime non-urban traffic. Therefore, a suitable design of these lighting systems is of the highest relevance to allow the driver an early detection of obstacles so that an appropriate reaction is possible. In order to provide a design basis for the headlamps, this article deals with the determination of the minimum required lighting conditions for reliable object detection. For this purpose, studies with drivers were conducted in a light tunnel on a closed test site, and the Visibility Level (VL) required for reliable object detection was considered. Gray cards with a reflectance of 4% were positioned on different positions of measurement grids, and the intensity of the low beam and high beam of a test vehicle was increased step by step until the drivers had detected the gray card. The results demonstrate that a Visibility Level of at least 13.35 is required in non-urban areas in order to reliably detect objects. In addition, the required Visibility Level depends on the eccentricity angle. Thus, the required Visibility Level reaches its maximum value in the foveal area of the field of view and decreases in a Gaussian shape in the periphery.

**Keywords:** automotive lighting; object detection; nighttime traffic; non-urban area; Visibility Level

## 1. Introduction

The visual system is the main source of information when driving a motor vehicle. It is used to detect potential risks at an early stage and to initiate an adjustment of the driving behavior [1–7]. Adequate visibility conditions are necessary for the visual system to perform its task reliably. To ensure these visibility conditions, especially in road traffic at night, motor vehicle headlamps are used. In this context, traffic in non-urban areas is of particular relevance, since vehicle headlamps are the only source of illumination that creates the detection conditions for the driver. Object detection describes the visual perception of luminance differences in the visual field. The smallest perceptible luminance difference between a visual sign ("object") and its environment is called the detection threshold. Thereby, the quantitative description of the detection conditions is made by different contrast definitions. The definition of the Weber contrast $C_W$, which describes the ratio of the luminance difference $\Delta L$ between object luminance $L_O$ and background luminance $L_B$ to background luminance $L_B$ (see Equation (1)), is of major relevance for nighttime road traffic [8].

$$C_W = \frac{L_O - L_B}{L_B} \tag{1}$$

Due to the large luminance range that can be used for both the object and the background, two contrast polarities can be realized, which are shown in Figure 1. Thus, positive contrast describes a situation where the object appears brighter than its environment ($L_O > L_B$). If the object appears darker than its background ($L_O < L_B$), it is called a negative contrast [8].

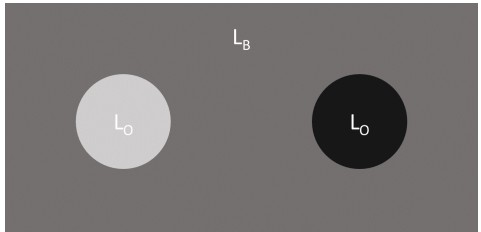

**Figure 1.** Contrast polarities; (**left**) The object appears brighter than its background (positive contrast); (**right**) The object appears darker than its background (negative contrast) [8].

Object detection depends on various parameters. In addition to the contrast polarity, the adaptation luminance, the object size, the observation time and the age of the observer influence the object detection [1,9–19].

Various research and analyses of traffic accident statistics demonstrate that the increased adaptation luminance provides a reduction in the required contrast through higher brightness levels, thus reducing the risk of nighttime traffic accidents [20–28]. For example, analyses by Scott [29] show that in the roadway luminance range of 0.5 to 2.0 cd m$^{-2}$, there is a direct correlation between roadway luminance and the night/day accident ratio, and an increase in the roadway luminance of 1.0 cd m$^{-2}$ provides a reduction in this ratio of about 35%. Moreover, studies by Damasky [18] or Blackwell [19] demonstrate similar results regarding the reduction of the required contrast at higher roadway or adaptation luminances.

Investigations on the influence of the object size of Aulhorn [15], Blackwell [19] or Schmidt–Clausen [30] on the required threshold luminance difference demonstrate that larger objects lead to a reduction of the threshold luminance difference. However, a distinction must be made between the Ricco's range, where object size has an influence, and the Weber's range, where the influence of object size is negligible [31,32].

The age influence on the detection of objects in road traffic at night could be shown by studies of Aulhorn [17], Blackwell and Blackwell [33], Schneider [34] and Weale [35]. Blackwell and Blackwell [33] performed a study with 235 observers of varying ages in this regard. Their analysis of the results for 234 of the 235 observers showed that the multiplier for the visibility threshold increases with age. The study was performed with 4 min Landolt rings on a background luminance of 100 cd m$^{-2}$. The observers had the task to indicate the recognition with a forced choice method. It is remarkable that the slope of the multiplier increases significantly from an age of 64 years (see Figure 2) [33].

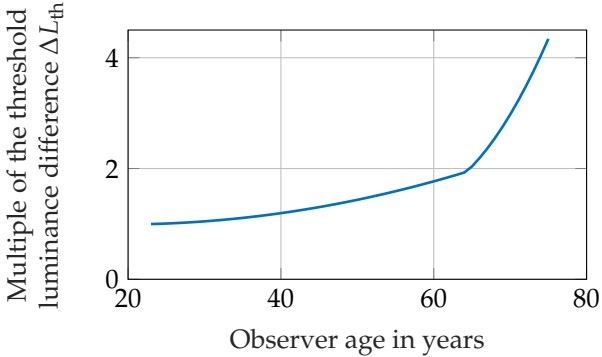

**Figure 2.** Age influence on the threshold luminance difference $\Delta L_{th}$ according to Blackwell and Blackwell [33]; the observer's age has a significant influence on the required threshold luminance difference $\Delta L_{th}$; especially from an age of 64 years, the age influence increases significantly [33].

Contrast polarity also affects the threshold luminance difference required for object detection [15,18,36]. Studies by Aulhorn [15] and Damasky [18] demonstrate that the threshold luminance difference for positive and negative contrast is of the same magnitude but lower for negative contrast than for positive contrast.

These influence parameters were transferred by different researchers and research groups into detection models, in order to allow a statement about the detectability of objects [31,37–39]. One of the most used models is the Small Target Visibility (STV) model by Adrian (see Equation (2)) [31]. The threshold luminance difference $\Delta L_{th}$ is calculated according to the following formula.

$$\Delta L_{\text{th}} = k \cdot \left( \frac{\sqrt{\Phi}}{\alpha} + \sqrt{L} \right)^2 \cdot \frac{a(\alpha, L_B) + t}{t} \cdot F_{CP} \cdot AF \qquad (2)$$

where

- $\Delta L_{\text{th}}$: Threshold luminance difference;
- $k$: Detection probability factor;
- $\left( \frac{\sqrt{\Phi}}{\alpha} + \sqrt{L} \right)^2$: Luminous flux and luminance function according to Ricco's and Weber's law;
- $\alpha$: Plane object size in angular minutes;
- $a(\alpha, L_B)$: Blondel-Rey constant;
- $t$: Observation time in seconds;
- $F_{\text{CP}}$: Contrast polarity factor;
- $AF$: Age factor.

This threshold luminance difference applies to a certain probability under laboratory conditions. To transfer this to the complexity of a real traffic situation, the Visibility Level VL is used as a multiplier. The Visibility Level is determined as the ratio of the currently prevailing luminance difference between the object and its background and the calculated threshold luminance difference (see Equation (3)).

$$VL = \frac{\Delta L}{\Delta L_{\text{th}}} \qquad (3)$$

To determine such a Visibility Level, Damasky [18] performed both laboratory and field tests. The studies were conducted on both a closed test site and in real road traffic at night. The results of the studies show that for each increase in the complexity level, for instance by the additional driving task, the locality (highway, country road, city) or the position of the objects, the detection conditions become more difficult. Thus, for the transition from the laboratory test to the static field test, a threshold contrast that is 12.7 times higher is needed in order to detect the objects. This factor is called the field factor and is comparable to the Visibility Level. Furthermore, Damasky confirms the dependence of the required thresholds on the object size in the field test, as shown in Figure 3 on the left side. On the right side of Figure 3, on the other hand, the dependence of the required object luminance $L_O$ on the eccentricity angle $\Theta$ is shown. To obtain the presented data, the data from the investigations of Damasky were mirrored at the eccentricity angle 0°. Here, it is noticeable that the dependence of the object luminance $L_O$ on the eccentricity angle $\Theta$ can be described by a Gaussian function and thus the required object luminance $L_O$ decreases with the increasing eccentricity angle $\Theta$ [18].

A similar field study was conducted by Schneider [34]. Here, the detection object was realized by a pedestrian, which was positioned at different eccentricity angles of $\Theta$ next to the roadway. The field study was conducted in two parts with older and younger drivers and a test vehicle with activated high beam headlamps. In the quasi-static test, the driver sat in the driver's seat and the object moved toward the test vehicle. The distance between the detection object and the center of the road did not change. In the dynamic case, the test person drove the vehicle along the test track at a speed of $80 \, \text{km} \, \text{h}^{-1}$. In both parts, the driver signalized the detection of the object by pressing a button and the detection distance was measured by GPS sensors. The results of the study demonstrate that older drivers have slightly shorter detection distances than younger drivers. Thus, the results regarding the negative effect of age on visual performance are confirmed. Furthermore, the

results demonstrate that the driving task also influences the detection distance. Thus, in the quasi-static part of the test, higher detection distances were achieved at all eccentricities versus in the dynamic case. The field factor, which, analogously to the Visibility Level, represents the multiplier between laboratory and field tests, is in a range of 8.75 to 32 for the entire test group. Thus, significantly higher contrasts are required in real traffic situations for the reliable detection of the visual signs than in the laboratory.

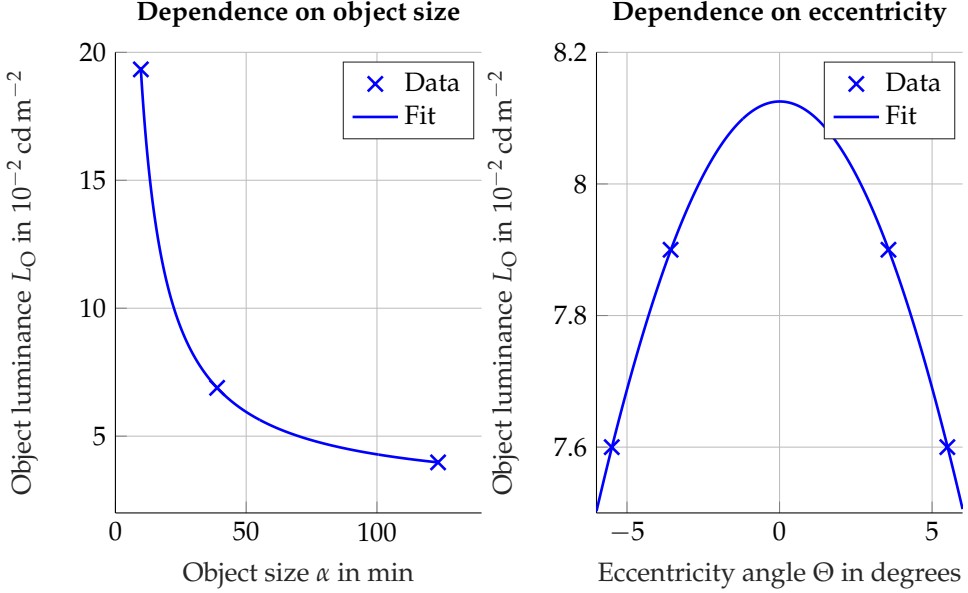

**Figure 3.** Dependence of the required object luminance $L_O$ on the object size $\alpha$ (**left**) and the eccentricity angle $\Theta$ (**right**) according to Damasky [18]; both with increasing object size $\alpha$ and with increasing eccentricity angle $\Theta$ the object luminance $L_O$ required for detection decreases; for both fit functions, the coefficient of determination $R^2$ is greater than 0.99 [18].

Based on the determined detection distances, which are on average about 90 m, no direct dependence of the detection distance on the eccentricity can be observed. Therefore, Schneider considered the contrasts required at the determined detection distances and calculated the required light intensity from this. Once again, the measured data was mirrored to take into account the object positions to the right and left of the roadway. The curve of the required light intensity for the detection object at a distance of approximately 90 m as a function of the horizontal object position is shown in Figure 4 [34].

Looking at the luminous intensity curve in Figure 4, it is evident that the luminous intensity required to detect a pedestrian at a distance of about 90 m decreases as the eccentricity angle $\Theta$ increases. Thus, the results from the field studies of Damasky [18] are qualitatively confirmed by the investigations of Schneider [34].

The studies conducted so far on the Visibility Level give an initial indication of the detection conditions required for nighttime road traffic in non-urban areas. However, dynamic field studies in particular have the disadvantage that the studies cannot be carried out reproducibly with several drivers and thus different conditions are present for each test run, which significantly influence the respective detection conditions. The present work is therefore intended to answer the following research questions in statically conducted field studies.

1. What Visibility Level is required for reliable object detection with a probability greater than 90 %?
2. What influence does the object's distance and angle have on the required Visibility Level?

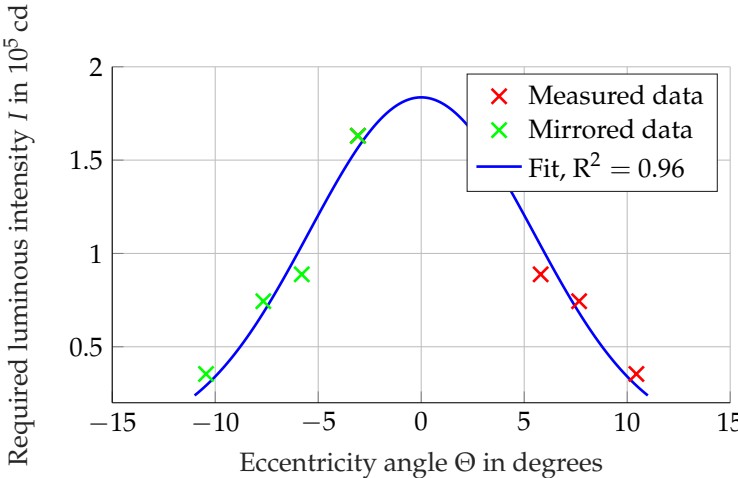

**Figure 4.** Required light intensity $I$ for a detection object at a distance of about 90 m as a function of the eccentricity angle $\Theta$ according to the field investigations of Schneider [34]; with increasing eccentricity angle, the required light intensity decreases, and the coefficient of determination of the correlation is $R^2 = 0.96$ [34].

## 2. Materials and Methods

The aim of the detection investigations in non-urban areas is to determine the minimum Visibility Level required for reliable object detection ($p \geq 90\%$) as a function of the object position. For this purpose, not only the distance dependence but also the angle dependence of object detection in nighttime road traffic is investigated. Furthermore, the general influence of environmental conditions on the Visibility Level required for detection is considered.

The study for object detection in non-urban areas was divided into two parts and conducted statically. Thus, on the one hand, the study took place under highly controlled conditions in a light tunnel and, on the other hand, the study was conducted on a closed test site. This separation was performed to determine the influence of distance and angle on object detection under controlled conditions. Conducting the study on the closed test site served the purpose of the more realistic replication of the traffic situation on rural roads.

A total of eleven drivers (age range: 18 to 59 years) participated in the study in the light tunnel, and 15 drivers (age range: 18 to 34 years) participated in the study on the closed test site. All drivers were in possession of a valid driving license at the time of the study and wore their visual aids when required. Thus, a visual acuity of at least 0.7 required for driving a motor vehicle was assumed and no additional visual acuity testing was performed.

To ensure that the performed studies represent worst-case scenarios, 20 cm × 20 cm large gray cards with a reflectance of about 4 % were used as detection objects following previous studies [31,40]. A low reflectance was chosen because, according to studies by Randrup Hansen and Schandel Larsen [41] or Schneider [42], the winter clothing of pedestrians, in particular, have low reflectance values of less than 10 %.

For the variation of the distance and the angle between the detection object and the observer, the gray card was set up on fixed positions of measuring grids. The measuring grids were thereby adapted to the available total dimensions of the respective experimental area. Thus, in the light tunnel, there was a usable area of 120 m × 12 m available, while a width of about 20 m was available on a roadway with a length of more than 1 km on the closed test site. Based on these limitations, the measurement grids, which can be observed in Figure 5, were designed for the partial studies.

The distances to the object rows of the measurement grids were 40 m, 60 m, 80 m, and 100 m for both studies. Due to the limitation of the light tunnel, as shown in Figure 5 on the left, a 4 × 5 measurement grid was feasible, with which a total of 20 object positions were

investigated during the detection study. At the closed test site, a total of 24 object positions were considered in a 4 × 6 measurement grid (see Figure 5 on the right). Tables 1 and 2 show the exact positions of the gray cards in front of the test vehicle in the respective measuring grids. Here, negative horizontal distances represent an offset to the left with respect to the longitudinal axis of the vehicle and positive horizontal distances represent an offset to the right with respect to the longitudinal axis of the vehicle.

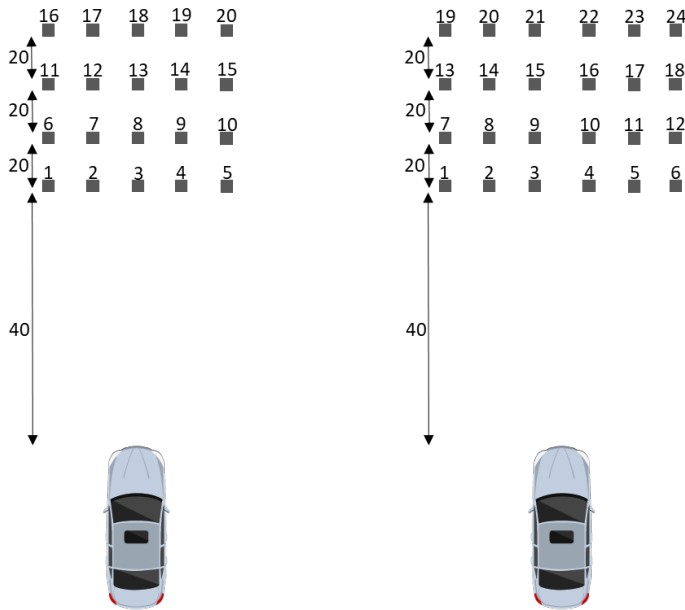

**Figure 5.** Schematic representation of the measurement grids for the detection studies in non-urban areas; in the light tunnel, a 4 × 5 measurement grid was realized (**left**), while on the closed test site a 4 × 6 measurement grid was achievable (**right**); the distances to the individual object rows were 40 m, 60 m, 80 m and 100 m; the numbers above the grid points indicate the internal designation of the respective object position.

Table 1 shows that in the light tunnel, a width of ±4.50 m was spanned by the measuring grid, where the horizontal distance between two grid points is 2.25 m. Furthermore, in each object row, there was one object on the longitudinal axis of the vehicle (offset: 0.00 m). Thus, at a constant angle, only the distance between the 40 m and 100 m was varied here.

**Table 1.** Assignment of object positions to measuring grid points in the light tunnel.

| Offset to Longitudinal | Distance to Vehicle in m | | | |
| Axis of Vehicle in m | 40 | 60 | 80 | 100 |
|---|---|---|---|---|
| −4.50 | Position 1 | Position 6 | Position 11 | Position 16 |
| −2.25 | Position 2 | Position 7 | Position 12 | Position 17 |
| 0.00 | Position 3 | Position 8 | Position 13 | Position 18 |
| 2.25 | Position 4 | Position 9 | Position 14 | Position 19 |
| 4.50 | Position 5 | Position 10 | Position 15 | Position 20 |

In contrast to the light tunnel, the total width on the closed test site was 18 m (±9.00 m) with horizontal distances between the grid points of 3.00 m. For this measurement grid, no object was positioned on the longitudinal axis of the vehicle, because here the gray cards would partially stand directly on road markings and thus influence the detection conditions at these positions.

**Table 2.** Assignment of object positions to measurement grid points on the closed test site.

| Offset to Longitudinal Axis of Vehicle in m | Distance to Vehicle in m | | | |
|---|---|---|---|---|
| | 40 | 60 | 80 | 100 |
| −9.00 | Position 1 | Position 7 | Position 13 | Position 19 |
| −6.00 | Position 2 | Position 8 | Position 14 | Position 20 |
| −3.00 | Position 3 | Position 9 | Position 15 | Position 21 |
| 3.00 | Position 4 | Position 10 | Position 16 | Position 22 |
| 6.00 | Position 5 | Position 11 | Position 17 | Position 23 |
| 9.00 | Position 6 | Position 12 | Position 18 | Position 24 |

A BMW 318d xDrive was used as test vehicle. The LED headlamps were modified in such a way that the dimming of the low beam and high beam was possible by means of an 8-bit PWM control (from 0 to 255). Thus, the absolute luminous intensity of the headlamps was adjusted via the PWM dimming and thereby the object luminance $L_O$ and the background luminance $L_B$ were varied. To evaluate the luminances, luminance images were recorded at the different PWM levels using a luminance measurement camera.

The study session started with an adaptation phase of about 15 min, during which the drivers received the instructions for the experimental procedure. Subsequently, the gray card was placed on the first randomly chosen position. The headlamp light intensity was increased in discrete PWM levels from 0 % to 100 % (8-bit values: 0 to 255), and the driver was tasked with signaling the detection of the gray card by pressing a button while fixating a second vehicle positioned behind the last row of objects in the driving direction. Gaze fixation was required to ensure that the angle between the observer and the detection object was reproducible for all drivers. After all PWM levels were completed at the current object position, the next object position was selected in a randomized order. When changing object positions, the driver was asked to close the eyes so that the next object position remained unknown at first. This procedure was repeated until the study was completed on all object positions. The study procedure was identical for the light tunnel and the closed test site. Figure 6 shows a schematic diagram of the test sequence for illustration purposes.

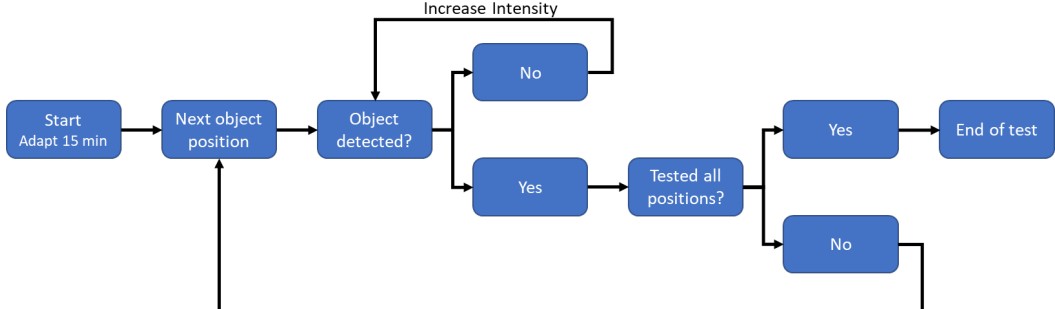

**Figure 6.** Flowchart of the experimental procedure for a driver.

The analysis of the recorded driver data was performed by the psychometric function according to Linschoten et al. [43] (see Equation (4)).

$$P(x) = \gamma + (1 - \gamma) \cdot \frac{1}{1 + \left(\dfrac{x}{\alpha}\right)^{-\beta}} \tag{4}$$

Here, $x$ represents the independent variable and $P(x)$ the associated probability of a positive response. This function is characterized by three parameters, where $\alpha$ describes the 50% threshold. The parameter $\beta$ describes the steepness of the curve and $\gamma$ indicates the probability that a positive response occurs purely by chance. Since this case is excluded

in the study conducted based on the stimuli presented, the value for $\gamma$ is set to 0 and the fit function is simplified, as shown in Equation (5).

$$P_{\gamma=0}(x) = \frac{1}{1 + \left(\dfrac{x}{\alpha}\right)^{-\beta}} \tag{5}$$

## 3. Results

In the following, the data evaluation based on the study in the light tunnel is described, since it is identical for both test environments. At the key points, the results for the closed test site are also discussed.

The data evaluation starts with the determination of the object luminance $L_O$ and the background luminance $L_B$ (mean value of the background luminances of the four surrounding areas) from the recorded luminance images, which are shown exemplarily for the light tunnel on the left side and for the closed test site on the right side in Figure 7. The luminance images are recorded in rows, so that one luminance image per PWM level is sufficient for the evaluation of all five or six object positions in the respective row.

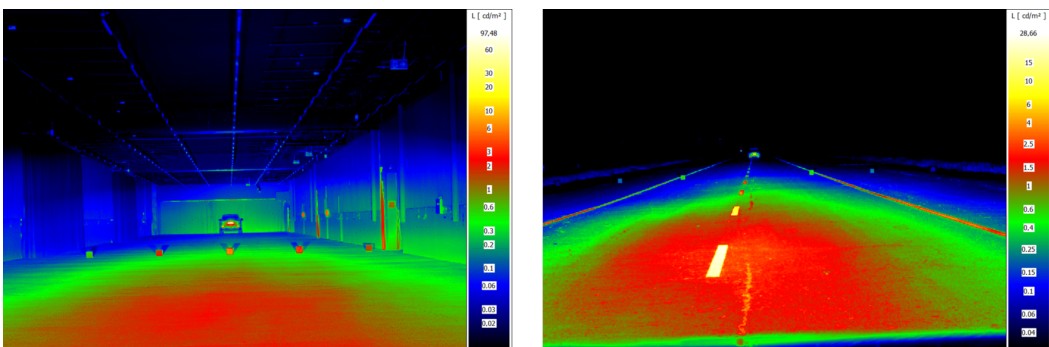

**Figure 7.** Exemplary representation of the recorded luminance images for the light tunnel (**left**) and the closed test site (**right**); the fixation vehicle, which is positioned behind the last row of objects in the driving direction, can be observed in both figures.

From the obtained data for object luminance $L_O$ and background luminance $L_B$, the luminance difference $\Delta L$ and Weber contrast $C_W$ are calculated by Equation (1). In addition, the threshold luminance difference $\Delta L_{th}$ is determined via the STV model (Equation (2)) and then the Visibility Level is determined using Equation (3).

The results of the photometric evaluation of the Weber contrast $C_W$ are shown in Figure 8 as an example for position 1, which is located at a distance of 40 m in front of the vehicle and a horizontal offset of $-4.50$ m from the longitudinal axis of the vehicle. From Figure 8, it can be observed that both the object luminance $L_O$ and the background luminance $L_B$ increase linearly from a PWM level of 45 (see Figure 8 on the left). This causes the luminance difference $\Delta L$ between the object and its environment to also increase linearly. Due to this linearity, the contrast curve, as shown in Figure 8 on the right, enters saturation with an increasing PWM level. This relationship can be described by calculating the Weber contrast $C_W$ with the linearly increasing luminances in Equation (6).

$$C_W(\text{PWM}) = \frac{L_O(\text{PWM}) - L_B(\text{PWM})}{L_B(\text{PWM})} = \frac{\Delta L(\text{PWM})}{L_B(\text{PWM})} = \frac{a \cdot \text{PWM}}{b \cdot \text{PWM}} = \frac{a}{b} = \text{const.} \tag{6}$$

As Equation (6) shows, the course of the luminance difference $\Delta L$ can be described as a multiple $a$ of the set PWM level. This is also the case for the background luminance $L_B$, which is linked to the set PWM level via the constant slope $b$. Thus, in the linear range of the luminance curve, a constant Weber contrast $C_W$ results, which can be represented as the ratio of the two slopes, $a$ and $b$.

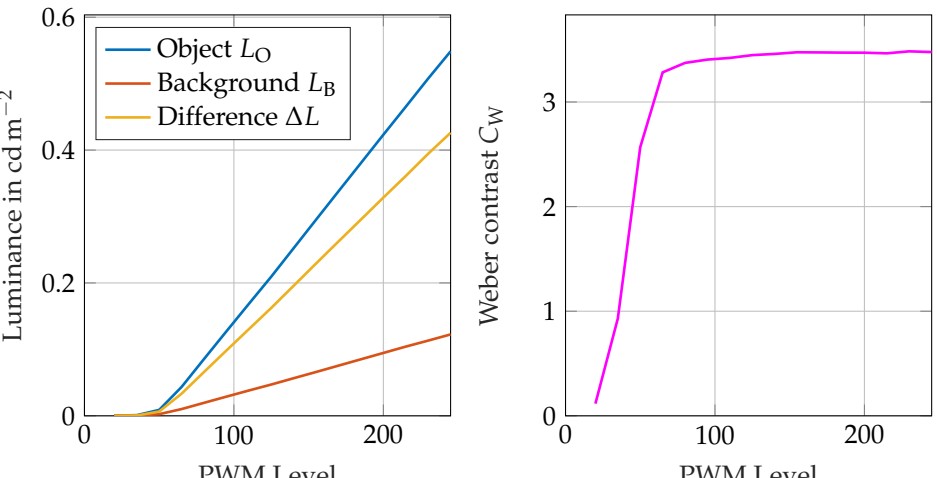

**Figure 8.** Luminances $L$ (**left**) and Weber contrast $C_W$ (**right**) of the gray card at object position 1 in the light tunnel; due to the linear increase in the luminance difference $\Delta L$ and the background luminance $L_B$, the contrast gradient enters saturation.

Since the Weber contrast $C_W$ enters a saturation range and thus does not allow for differentiation at higher PWM levels, the Visibility Level shown in Figure 9 on the right is used for further evaluation. As Figure 9 shows, the Visibility Level increases with the increasing PWM level and thus enables a differentiated view over the entire adjustable intensity range of the LED headlamps of the test vehicle in the further evaluation.

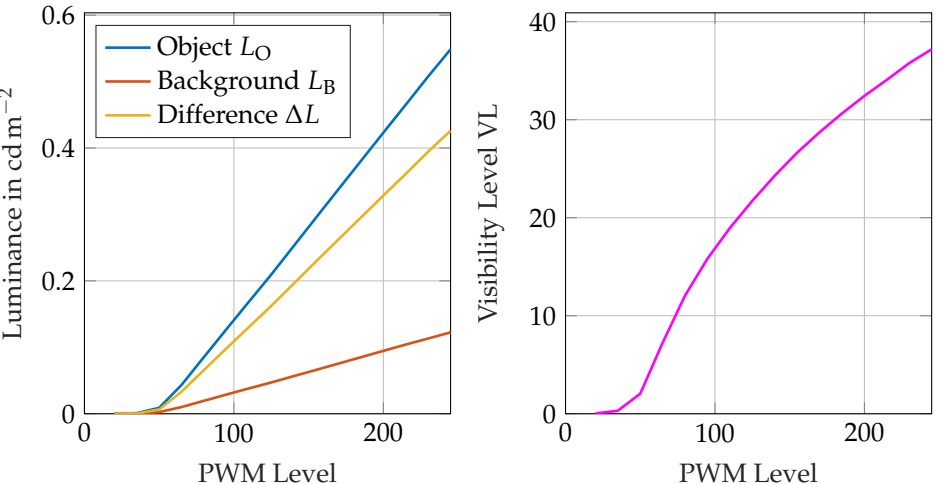

**Figure 9.** Luminance $L$ (**left**) and Visibility Level VL (**right**) of the gray card at object position 1 in the light tunnel; both luminance and Visibility Level increase with higher PWM levels.

Thus, the Visibility Level generated by the various control of the LED headlights at the different object positions represents the independent variable. The dependent variable is the detection probability of the gray card at a given Visibility Level. This is calculated by determining the proportion of drivers who detected the gray card at the respective Visibility Level. Since no random positive responses are to be expected, the correlation is calculated with the simplified psychometric function according to LINSCHOTEN et al. [43] from Equation (5). In the context of the present work, reliable object detection is considered to occur at a detection probability of 90 %. Other relevant thresholds are the 50 % threshold ("critical threshold") and the 70 % threshold ("passable detection").

The evaluation of the object detection probability with the psychometric function is shown in Figure 10 for object position 1 in the light tunnel. For all other object positions, the evaluation is performed in the same way.

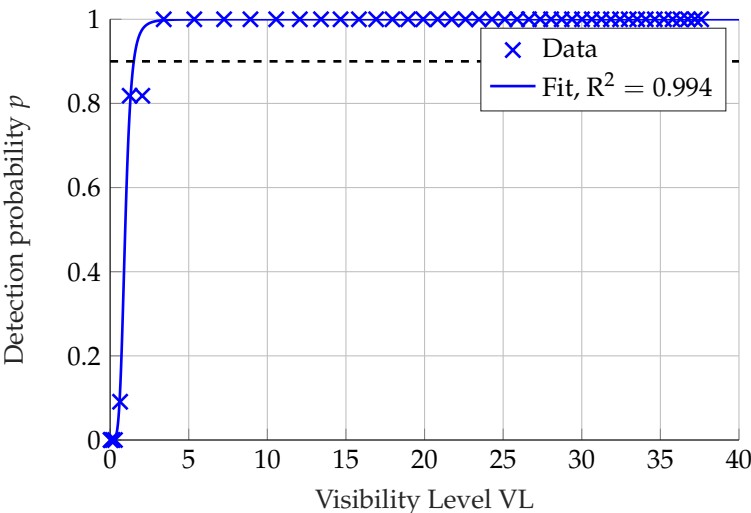

**Figure 10.** Relationship between the detection probability *p* and the Visibility Level VL of the gray card at object position 1 in the light tunnel; to achieve a detection probability *p* of 90 %, a Visibility Level of 1.505 is required.

Figure 10 shows that the relationship between the existing Visibility Level and the detection probability is very well represented by the psychometric function ($R^2 = 0.994$). If the detection probability of the gray card at position 1 shown in Figure 10 is considered, it can be observed that from a Visibility Level of 1.505, the detection probability is at least 90 % and is thus called "reliably detected". In this way, all further object positions are evaluated, and a spatially resolved distribution of the required Visibility Level is determined, which is shown in Table 3.

**Table 3.** Required Visibility Level on the different object positions in the light tunnel.

| Offset to Longitudinal | Distance to Vehicle in m | | | |
|---|---|---|---|---|
| Axis of Vehicle in m | 40 | 60 | 80 | 100 |
| −4.50 | 1.505 | 2.029 | 2.480 | 3.454 |
| −2.25 | 2.384 | 2.173 | 6.380 | 4.804 |
| 0.00 | 7.760 | 6.295 | 21.500 | 9.920 |
| 2.25 | 3.252 | 5.140 | 4.300 | 6.610 |
| 4.50 | 10.955 | 8.220 | 7.450 | 15.030 |

From Table 3, it can be observed that the required Visibility Level ranges from 1.505 to 21.500. The value of 21.500 represents an outlier resulting from the proximity of the object position to the fixation object. Looking at the measurement series at the individual distances, it is striking that the required Visibility Level decreases at the transition from the foveal visual field (offset: 0.00 m) to the peripheral visual field. An exception is the horizontal offset of 4.50 m to the right. Here, the Visibility Level increases again. As stated by the drivers, the potential cause for this is the proximity of the objects to the wall of the light tunnel (see Figure 7 on the left). Thus, the drivers indicate that this proximity to the wall of the light tunnel makes the detection of the objects significantly more difficult. For this reason, the object positions on the far right are not considered further when considering the angle dependence.

The angle dependence of the Visibility Level is determined after removing these data points. For this purpose, the eccentricity angles to the respective measurement grid points are determined, and the data for the different distances are summarized. The summarization of the data is made possible by the fact that the distance is already taken

into account in the calculation of the threshold luminance difference $\Delta L_{\text{th}}$ with the STV model and the object size $\alpha$ required for this purpose (see Equation (7)).

$$\alpha = arctan\left(\frac{D}{d}\right) \tag{7}$$

Here, $D$ describes the extension of the object in m and $d$ the distance to the test vehicle in m. The result of this evaluation is shown in Figure 11. To describe the correlation between the eccentricity angle $\Theta$ and the required Visibility Level, a Gaussian function with two terms after Equation (8) is used due to the data set.

$$f(x) = a_1 \cdot exp\left(-\left(\frac{x - b_1}{c_1}\right)^2\right) + a_2 \cdot exp\left(-\left(\frac{x - b_2}{c_2}\right)^2\right) \tag{8}$$

Here, $x$ describes the independent variable and $f(x)$ the angle-dependent Visibility Level. $a_1$ and $a_2$ are the maximum values of the respective Gaussian functions. Via $b_1$ and $b_2$, the horizontal displacement of the Gaussian functions related to the eccentricity angle of 0° is described. $c_1$ and $c_2$ define the width of the respective Gaussian functions. The evaluation of the data with this equation shows that there is a Gaussian correlation between the eccentricity angle $\Theta$ and the Visibility Level, which is required for reliable object detection. Thus, the findings from previous studies by Damasky [18] and Schneider [34] are confirmed by the study conducted in the present work.

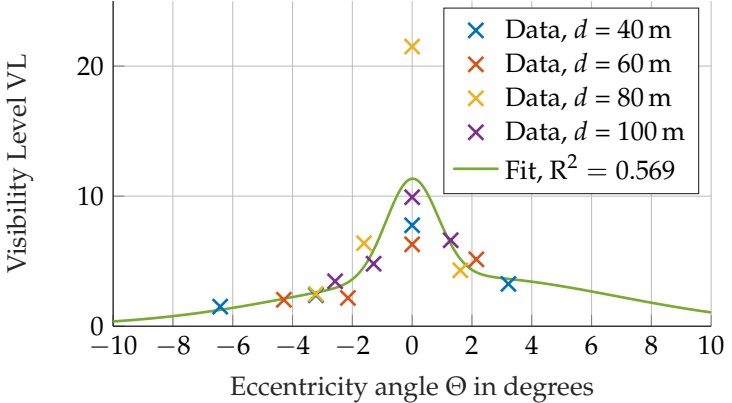

**Figure 11.** Angular dependence of the Visibility Level VL; as the eccentricity angle $\Theta$ increases, the required Visibility Level for reliable object detection decreases.

The angle dependence of the Visibility Level for reliable object detection shown in Figure 11 results from the distribution of the receptors (cones and rods) on the retina [44–47]. Due to the fact that the examinations were performed in the mesopic range typical for nighttime road traffic, the rods are activated in addition to the cones by the adaptation time of 15 min. Since the rods, which are more light-sensitive than the foveally concentrated cones, are concentrated in the peripheral area of the retina, the brightness sensitivity of mesopic vision is higher in the peripheral area and the contrast perception is improved compared to the foveal vision [44–47].

The Gaussian relationship between the eccentricity angle $\Theta$ and the required Visibility Level shown in Figure 11 shows that for the design of optimized luminous intensity distributions, the foveal area represents the critical area ("worst-case"). Thus, the luminous intensity must be determined on the basis of the required Visibility Level for the foveal area and then adopted for the peripheral area.

In order to determine the required Visibility Level over the different object positions and thus create a common basis for the design of the headlamp light distributions, the relationship between the proportion of detected object positions and the Visibility Level is

considered. For this consideration, the object positions are included again at an offset of 4.50 m, and thus all 20 object positions in the light tunnel are analyzed.

The detected object positions are evaluated using the calculated Visibility Levels in Table 3. Thus, the proportion of detected object positions is determined by summing, at the respective Visibility Level, the number of object positions at which the gray card is detected with a probability *p* of at least 90 %. While at a Visibility Level of 1.505, only the gray card is detected on the object position 1; at a Visibility Level of 5.140, there are already 10 object positions on which the gray card is detected with a probability of 90 %. In order to detect the gray card on all object positions considered, a Visibility Level of 21.500 is necessary.

This procedure yields the Visibility Level as an independent variable and the proportion of object positions as a dependent variable. To test these two variables for correlation, again, the simplified psychometric function of Linschoten et al. [43] is used (Equation (5)). The result of this analysis is shown in Figure 12 and indicates, with a coefficient of the determination of $R^2 = 0.984$, that there is a correlation between the Visibility Level and the proportion of object positions with a detection probability of the gray card of at least 90 %.

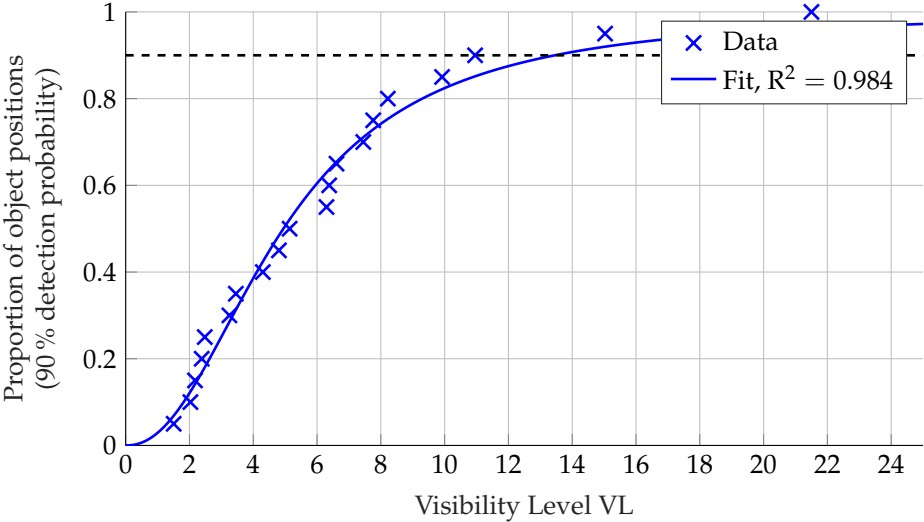

**Figure 12.** Relationship between the Visibility Level VL and the proportion of object positions in the light tunnel at which the gray card is reliably detected; to reliably detect the gray card at 90% of the object positions, a Visibility Level of 13.35 is required.

From Figure 12, it can be observed that a Visibility Level of 13.35 is required for the reliable detection of the gray card on 90% of the object positions. If a Visibility Level of about 4.95 is provided to the vehicle driver, a reliable detection of the gray card on 50% of the object positions is achieved. In this context, the 50% threshold represents the critical threshold for object detection in nighttime road traffic, since the number of detections and the number of missed detections are equal here.

In order to analyze the influence of the environmental conditions, the results of the investigation on the closed test site are presented here. For this purpose, the Visibility Level required at the various object positions for reliable object detection is first considered in Table 4.

Table 4 shows that the required Visibility Level is in the range of 2.585 to 25.439 and thus higher Visibility Levels are required for the reliable detection on the closed test site, which is a realistic representation of nighttime road traffic. To consider the influence of the environment in more detail, the results for the closed test site are also analyzed using the simplified psychometric function. Figure 13 shows the relationship between the Visibility Level and the proportion of object positions on which the gray card is reliably detected.

**Table 4.** Required Visibility Level on the different object positions on the closed test site.

| Offset to Longitudinal | Distance to Vehicle in m | | | |
|---|---|---|---|---|
| Axis of Vehicle in m | 40 | 60 | 80 | 100 |
| −9.00 | 2.585 | 25.439 | 22.967 | 20.100 |
| −6.00 | 19.58 | 24.755 | 5.219 | 10.879 |
| −3.00 | 5.000 | 13.215 | 6.772 | 20.119 |
| 3.00 | 5.750 | 12.670 | 4.893 | 6.243 |
| 6.00 | 4.715 | 14.960 | 6.052 | 7.746 |
| 9.00 | 4.453 | 14.434 | 4.817 | 11.341 |

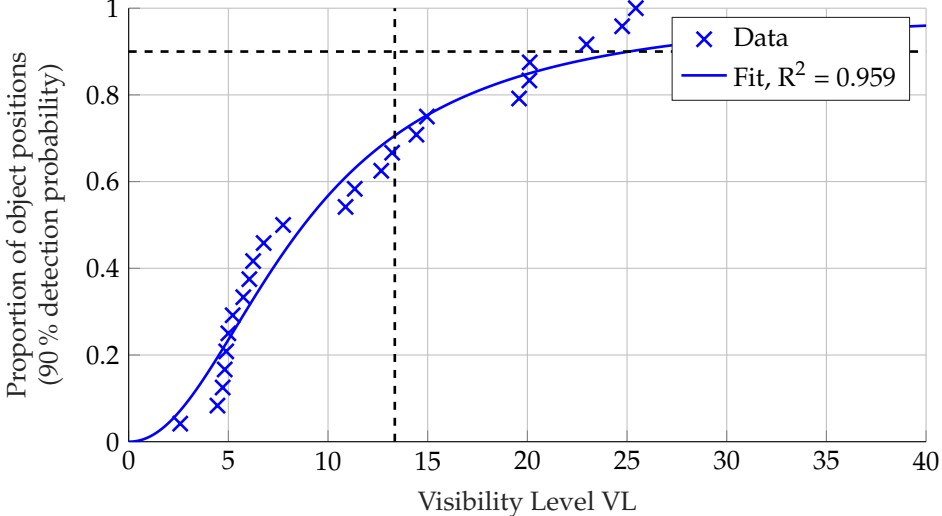

**Figure 13.** Relationship between the Visibility Level VL and the proportion of object positions on the closed test site at which the gray card is reliably detected; while a Visibility Level of 25 is required to reliably detect the gray card at 90% of the object positions, at the Visibility Level of 13.35 determined in the light tunnel, the gray card is detected at more than 70% of the object positions.

As Figure 13 shows, the relationship between the Visibility Level and the proportion of reliably detectable object positions on the closed test site can also be described via the psychometric function ($R^2 = 0.959$). The increase in the required Visibility Level for the reliable detection of the gray card to 90% of the object positions from Table 4 is confirmed by considering the psychometric function. Thus, at the closed test site, a Visibility Level of about 25 is required to reliably detect the gray card at 90% of the object positions. This corresponds to an increase in the Visibility Level by a factor of 1.87 compared to the light tunnel. In addition to the 90% threshold, the Visibility Level of 13.35 from the investigation in the light tunnel is indicated by the vertical dashed line in Figure 13. Looking at the percentage of reliably detectable object positions at the Visibility Level of 13.35, it can be observed that the gray card is reliably detected on more than 70% of the object positions. Thus, the critical 50% threshold, which is reached at a Visibility Level of about 8.8, is exceeded by using the Visibility Level determined in the light tunnel.

For the design of motor vehicle headlamp light distributions, a Visibility Level of 13.35 must therefore be guaranteed as an absolute minimum requirement in order to produce passable detection conditions ($p > 70\%$) for vehicle drivers in nighttime road traffic in non-urban areas.

## 4. Discussion

This section reviews the studies on object detection in road traffic at night in non-urban areas by addressing and answering the research questions derived in the introduction of this article.

1.  What Visibility Level is required for reliable object detection with a probability greater than 90%?

    The studies carried out demonstrate that the required Visibility Level for reliable object detection depends on the ambient conditions. Thus, in non-urban areas under controlled ambient conditions in the light tunnel, a Visibility Level of 13.35 is required to detect the gray card with a reflectance of about 4% on 90% of the considered object positions with a probability of 90%. This value is comparable to the field factor of 12.7 that Damasky [18] determined in his laboratory and field studies.

    On the closed test site in non-urban areas, the required Visibility Level for reliable object detection on 90% of the object positions increases to 25. This is due to the environmental conditions, which are not as strictly controllable as in the light tunnel and thus create more complex detection situations. This confirms the findings of Adrian [12,13,31], Damasky [18], Bacelar [48,49] and also Schneider [34], in which the required luminance difference increases with increasing complexity of the traffic situation; thus, the definition of a Visibility Level for such situations turns out to be useful. Nevertheless, the Visibility Level of 13.35 determined in the light tunnel ensures a reliable detection of the gray card at more than 70 percent of the object positions on the closed test site.

2.  What influence does the object's distance and angle have on the required Visibility Level?

    In order to determine the angle and distance dependence of the required Visibility Level, strictly controllable conditions, such as those found in the light tunnel, are necessary. The detection study carried out in the light tunnel shows that the Visibility Level required for reliable object detection has a Gaussian shape. The maximum of the Gaussian shape of the Visibility Level is in the foveal area around 0°. This correlation, which has been demonstrated by studies of Damasky [18] and Schneider [34], can be explained by the distribution of rods and cones on the human retina. Thus, the higher density of the more light-sensitive rods in the peripheral visual field provides better contrast perception, which is not available in the foveal area due to the higher cone density [44–47]. Since the distance to the object is already taken into account by determining the object size in angular minutes, the distance dependence is already included in the calculated Visibility Level. An additional distance dependence of the required Visibility Level cannot be extracted from the study results.

## 5. Conclusions and Outlook

The presented article deals with the determination of necessary Visibility Levels VL in nighttime non-urban road traffic in order to reliably detect objects. For this purpose, detection studies are carried out in a light tunnel and on a closed test site. Gray cards with a reflectance of about 4% are used as detection objects, which are positioned in a measurement grid in front of the vehicle to determine object detection as a function of the distance and eccentricity angle. For this purpose, object distances from 40 m to 100 m are considered. Therefore, both the low beam and high beam intensities are varied in the detection studies, while the drivers are tasked with signaling the object detection.

The results of the light tunnel study demonstrate that a Visibility Level of 13.35 is required for a detection probability of 90%. This value is comparable to the field factor of 12.7 determined by Damasky [18]. If the more realistic situation on the closed test site is considered, the required Visibility Level increases to 25. Furthermore, the angle dependence of the required Visibility Level can be observed in the light tunnel. This follows a Gaussian course and shows the maximum required Visibility Level in the foveal area (0°). This course confirms the results of the investigations of Damasky [18] and Schneider [34].

In order to further increase the validity of the conducted studies in the future, the following extension options should be considered. The gray cards with a reflectance of about 4% used in the conducted studies represent a worst-case consideration. For future studies, it is therefore advisable to vary the object size, object shape, and reflectance and thus investigate their influence on the detection conditions. Unfortunately, due to

the pandemic situation, the number of drivers participating in the study is rather small. Therefore, the driver collective should be expanded, especially with drivers of a higher age, in order to specifically investigate the influence of observer age on the object detection probability. Furthermore, the results of the present study demonstrate that the eccentricity angle between the object and the driver has a significant influence on the required Visibility Level for a reliable object detection. Thus, the driver's gaze behavior should also be taken into account in future studies.

The results of the presented article can be used to optimize future vehicle headlamp light distributions to ensure the reliable object detection in nighttime non-urban traffic. Segmented light distributions, whose luminous intensities are designed on the basis of the required Visibility Level, could thus be implemented. This would have the advantage that, on the one hand, the object detection is ensured by sufficient luminous intensity and, on the other hand, the glare for other road users can be minimized by switching off the respective segments, as required. This type of adaptive lighting control is already being implemented under the name of a glare-free high beam and represents the state of the art [50,51].

Within the context of the study carried out, static tests are performed because they guarantee the reproducibility of the study and, consequently, the results obtained compared to dynamic tests, which are very suitable for the validation of light distributions. Due to the greater realism of dynamic studies, they should be performed as a next step to validate the light distributions generated by the static tests. Such studies can also be used to determine the influence of the driving task on the object detection probability.

**Author Contributions:** Conceptualization, A.E., D.H. and N.K.; Data curation, A.E., N.K. and T.V.; Formal analysis, A.E. and N.K.; Methodology, A.E., D.H. and N.K.; Software, A.E., D.H., N.K. and T.V.; Supervision, T.Q.K.; Validation, A.E.; Visualization, A.E.; Writing—original draft, A.E.; Writing—review and editing, A.E., D.H., N.K., T.V., K.K., M.A.P. and T.Q.K.; Project administration, A.E.; Funding acquisition, A.E., D.H. and T.Q.K. All authors have read and agreed to the published version of the manuscript.

**Funding:** This research was funded by the Deutsche Forschungsgemeinschaft (DFG, German Research Foundation) under grant no. 450942921. We further acknowledge support by the Open Access Publishing Fund of the Technical University of Darmstadt.

**Institutional Review Board Statement:** Not applicable.

**Informed Consent Statement:** Not applicable.

**Data Availability Statement:** All data generated or analyzed to support the findings of the present study are included in this article. The raw data can be obtained from the authors, upon reasonable request.

**Acknowledgments:** This article incorporates the results of the doctoral thesis submitted in 2023 to the Laboratory of Adaptive Lighting Systems and Visual Processing at the Technical University of Darmstadt, Germany; with the title "Optimierung von Frontscheinwerferlichtverteilungen anhand wahrnehmungsphysiologischer Kriterien" (English: "Optimization of headlamp light distributions based on physiological perception criteria") by Anil Erkan.

**Conflicts of Interest:** The authors declare no conflict of interest.

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
