# Peer review of "Required Visibility Level for Reliable Object Detection during Nighttime Road Traffic in Non-Urban Areas"

_applsci, doi:10.3390/app13052964_

Round 1

Reviewer 1 Report

Dear Editors, Dear Authors

Thank you for the opportunity to review this article.

I think that the article is a valuable source of information helpful in making progress in vehicle headlight designing. I am impressed by the scope of the collected material and the analysis carried out and I think that the subject matter of the article fits well in the domain of the journal. The purpose and scope of the work was formulated correctly and the methodology was matched appropriate for that type of research. The scope and size of the sample data, as well as the statistical methods used, made it possible to elaborate the results and draw the adequate conclusions.

However, I have found some imperfections that should be improved.

The methodology could be described better.

The past tense should be used when describing individual research activities and it should not be confused it with the present tense.

I understand that "the subjects" are the people taking part in the research and that they are drivers, so it would be better to define them as drivers.

It is not clear how the road tests were carried out. What were the locations of the objects along the 1 km road and how the distance measurements were carried out while driving. How was the driver's perception of the object recorded? The description of these studies is very poor.

The chapter “discussion” should be reorganized and the results of the research should be compared to other similar research results. The chapter "Conclusions" contains many repetitions with respect to the previous text.

Perhaps it would be possible to present a broader summary of the research carried out. It is worth mentioning what should be paid attention to in such research, what should be tested, what corrections should be made to the research, what should be avoided? The lines - 419 - 430 – contain a suitable summary that can be extended.

After minor corrections, I recommend the article for publishing.

Some detailed comments are given below:

Line 77 – it should be precise in what kind of angular measure the object’s size is expressed - plane or solid, and if in the plane it could be added that only one dimension matters.

Line 110 – There can be the dot/point (80 km×h-1) as it is in the line 163 or 171.

Line 167 – full stop at the end of the sentence is missing.

Lines 201- 215 - The grammatical tense should be in the past tense, since the research had already been done. The description of the activities should be corrected.

Equation 4 - a was already used, so it could be used  a different symbol. It isn’t explained what is P(x) and x

Line 240 – what is the unit of PWM?

Figure 7 – the description of the vertical axis of the right plot is too close to the left plot.

Figure 7 and Figure 8 – the description of horizontal axis – accordingly to line 204 the PWM ranged from 0 to 100 %, so how can the axis be up to 250 (what are the units).

Lines 287 – 288 – please correct the language style.

Equation 7 - a - is this the same a?

Line 294 – the end of the line – test vehicle – which one is that? The one with the examined driver or the one being followed.

Equation 8 – There is no explanation of; f(x), x, b1, b2, c1, c2,

Line 304 – not “examinations are” but  “examinations were”

Kind regards,

Reviewer 2 Report

The manuscript addresses the minimum visibility level that drivers need in nighttime driving under a static approach when headlights are the only source of lighting.

In general, the manuscript has good readability, the literature review is complete and relevant, the experiment is clearly explained and discussed, and the conclusions are supported by the results. The authors acknowledge some limitations, which is a valuable exercise of honesty. I consider that the authors should address some issues before their manuscript is suitable for publication:

Line 6: remove abbreviation or include between parentheses.

Line 105: the authors refer to the effect of eccentricity angle on object detection in the literature review. However, this variable, although present in the experiment, was not considered by their analysis.

Line 135: the authors endorse their experimental research based on the lack of repeatability of visibility tests under dynamic conditions. However, dynamic visibility tests reproduce better the real visibility factors while driving. See:

Gibbons, R.B.; Medina Flintsch, A.; Williams, B.; Du, J.; Rakha, H. (2012). Sag Vertical Curve Design Criteria for Headlight Sight Distance; Blacksburg, VA.

Line 158 and 159. The number of subjects is too scarce to achieve statistically significant results.

Assuming that visibility level is expressed as a percentage, the axis labels should indicate it where applicable.

Line 378: Is it possible to supply a VL of 13.35 while avoiding glare taking into account that road geometry affects both visibility and headlight glare? Please discuss. See:

Alcón Gil, P.; De Santos-Berbel, C.; Castro, M. (2021) Driver Glare Exposure with Different Vehicle Frontlighting Systems. J. Safety Res., 76, 228–237, https://doi.org/10.1016/j.jsr.2020.12.018.

Akashi, Y., Hu, F., & Bullough, J.D. (2008). Sensitivity analysis of headlamp parameters affecting visibility and glare. (No. DOT HS 811 055). Springfield, VA.

Van Derlofske, J., Bullough, J.D., Dee, P., Chen, J., & Akashi, Y. (2004). Headlamp parameters and glare. SAE Technical Papers 2004-01-12. https://doi.org/10.4271/2004-01-1280.

Round 2

Reviewer 2 Report

After having read the response to the reviewers’ comments, I exhort the authors to make more effort when addressing the criticisms. The authors must indicate the lines of the last version of the manuscript where they have incorporated the changes addressing the reviewers’ comments.

In response to my comment concerning the repeatability of visibility tests, I have not found where in the text the authors inserted the corresponding text (they should have done it).

In response to my comment concerning the scarce number of participants, I have not found where in the text the authors addressed this limitation. It must be at least acknowledged in the conclusions.

Line 447. It is unclear what the authors mean by “glare for other road users can be minimized by switching off the respective segments as required.”
